# Modeling the impact of neuromorphological alterations in Down syndrome on fast neural oscillations

**Pau Clusella** [1]*, **Linus Manubens-Gil** [2]*, **Jordi Garcia-Ojalvo** [3], **Mara Dierssen** [3,4,5,6]

**1** Department of Mathematics, Universitat Politècnica de Catalunya, Manresa, Spain, **2** New Cornerstone Science Laboratory, SEU-ALLEN Joint Center, Institute for Brain and Intelligence, Southeast University, Nanjing, Jiangsu, China, **3** Department of Medicine and Life Sciences, Universitat Pompeu Fabra, Barcelona, Spain, **4** Centre for Genomic Regulation (CRG), The Barcelona Institute for Science and Technology (BIST), Barcelona, Spain, **5** Systems Neurology and Neurotherapies, Hospital del Mar Research Institute, Barcelona, Spain, **6** Center for Biomedical Research in the Network of Rare Diseases (CIBERER), Spain

* pau.clusella@upc.edu (PC); linusmg@seu.edu.cn (LM-G)

**Data Availability Statement:** All code and data in support of this publication are publicly available at https://github.com/pclus/neuromorphology.

## Abstract

Cognitive disorders, including Down syndrome (DS), present significant morphological alterations in neuron architectural complexity. However, the relationship between neuromorphological alterations and impaired brain function is not fully understood. To address this gap, we propose a novel computational model that accounts for the observed cell deformations in DS. The model consists of a cross-sectional layer of the mouse motor cortex, composed of 3000 neurons. The network connectivity is obtained by accounting explicitly for two single-neuron morphological parameters: the mean dendritic tree radius and the spine density in excitatory pyramidal cells. We obtained these values by fitting reconstructed neuron data corresponding to three mouse models: wild-type (WT), transgenic (TgDyrk1A), and trisomic (Ts65Dn). Our findings reveal a dynamic interplay between pyramidal and fast-spiking interneurons leading to the emergence of gamma activity ($\sim 40$ Hz). In the DS models this gamma activity is diminished, corroborating experimental observations and validating our computational methodology. We further explore the impact of disrupted excitation-inhibition balance by mimicking the reduction recurrent inhibition present in DS. In this case, gamma power exhibits variable responses as a function of the external input to the network. Finally, we perform a numerical exploration of the morphological parameter space, unveiling the direct influence of each structural parameter on gamma frequency and power. Our research demonstrates a clear link between changes in morphology and the disruption of gamma oscillations in DS. This work underscores the potential of computational modeling to elucidate the relationship between neuron architecture and brain function, and ultimately improve our understanding of cognitive disorders.

## Author summary

The structural integrity of individual brain neurons and the intricate networks they form are fundamental to all brain functions, with structural anomalies directly linked to

**Funding:** PC and JGO have received funding from the Future and Emerging Technologies Programme (FET) of the European Union's Horizon 2020 research and innovation programme (project NEUROTWIN, grant agreement No 101017716). JGO was also financially supported by the Spanish Ministry of Science and Innovation (project reference PID2021-127311NB-I00), the FEDER funds of the Spanish State Research Agency (project reference MICIN/AEI/10.13039/501100011033/FEDER), the Maria de Maeztu Programme for Units of Excellence in R\&D of the Spanish State Research Agency (project reference CEX2018-000792-M), and by the ICREA Academia program. LMG was supported by the National Natural Science Foundation of China (RFIS-I grant, reference 32350410413). MD acknowledges financial support from the Spanish Ministry of Science and Innovation (projects CPP2022-009659 and RTC2019-007329-1), the Spanish State Research Agency (project INTO-DS, reference PID2022-1419000B-I00), the Marato-TV3 Foundation (202212-30-31-32), and the European Comission Horizon 2020 programme (projects H2020-899986 and GO-DS21-848077). The funders had no role in study design, data collection and analysis, decision to publish, or preparation of the manuscript.

**Competing interests:** The authors have declared that no competing interests exist.

neurological disorders. Deciphering these links is a leading question in developmental disorders such as Down syndrome. Our work sheds light on the pivotal role that the structure of complex neural systems plays in shaping emergent network activity. In particular, our anatomically informed network modeling enables determining the extent to which specific deficits in neuronal architecture and connectivity perturb oscillatory patterns of activity.

## Introduction

Down Syndrome (DS), caused by the trisomy of chromosome 21, is associated with a wide spectrum of cognitive deficits [1], making it the most prevalent form of intellectual disability. Notably, abnormalities in the nervous system of individuals with DS manifest already at the single-neuron level. Mouse models of DS exhibit a significant reduction of dendritic tree branching and spine density when compared to control groups [2–4], features that have also been found in human postmortem tissue [5]. These morphological alterations are believed to play a significant role in the disruption of neural circuitry, ultimately contributing to the cognitive impairments associated with DS. Nevertheless, the precise mechanisms underlying the relationship between microscopic morphological alterations and mesoscopic brain dysfunction remain unknown.

Electrophysiological studies also show abnormal neural synchronicity in DS [6]. In particular, gamma rhythms ($\sim$ 40 Hz) appear to be significantly reduced in both awake and anesthetized DS mouse models [7]. Alterations of these fast neural rhythms are not exclusive to DS, and have also been observed in other neuropathologies, such as Alzheimer's disease [8, 9]. Gamma oscillations emerge from the collective activity of neural networks in both the hippocampus and cortex, and have been consistently associated with various cognitive processes, such as decision-making and memory tasks, across different species [10–17].

Thus we hypothesized that there is a direct link between microscopic circuitry abnormalities and functional deficits. However, this relation cannot be tested experimentally due to the presence of confounding factors in animal models. For instance, Ruiz-Mejias et al. [7] also reported a significant reduction of inhibitory connections targeting parvalbumin-positive interneurons in DS. This weakened recurrent inhibition was hinted as a potential cause for the reduction of gamma oscillations using a computational model, but the role of neuromorphological alterations was not explored in that study.

Here we propose a data-driven computational model of a simplified local neural network that incorporates some of the observed neuromorphological changes present in DS mouse models. We selected two different mouse models. The first model is trisomic for about two-thirds of the genes orthologous to human chromosome 21, (Ts(17(16))65Dn) [18], and is a well-characterized model for studying DS. We refer to this genotype as Ts65Dn for brevity. The second model (TgDyrk1A) [19], overexpresses only the dual-specificity tyrosine phosphorylation-regulated kinase 1a (*Dyrk1a*), a gene whose overexpression recapitulates the main neuronal architecture defects and cognitive impairments of the trisomy [20]. By integrating empirical data on dendritic complexity and spine density obtained from wild-type (WT), transgenic (TgDyrk1A), and trisomic mice (Ts65Dn), we construct a simplified cortical-layer structural model representing the synaptic connectivity of a neural network composed of point neurons, including pyramidal and fast-spiking interneurons. Simulations of these neural networks using Izhikevich dynamics [21] provide the functional differences between the three genotypes. Specifically, the model reproduces the deficit in gamma oscillations observed in DS animal models and allows us to test the role of reduced recurrent inhibition observed in [7].

Moreover, the scalable nature of the modeled morphologies enables us to explore the morphological space. This exploration includes values that do not correspond to specific animal models, allowing us to assess the impact of fabricated topologies on gamma rhythm generation.

Altogether, our study offers new insights into the complex interplay between neural morphology and network-level dynamics, thus contributing to our understanding of neurodevelopmental disorders.

## Results

Our model consists of an *in silico* representation of a cross-section of layers II/III of the mouse motor cortex. In this neural network, we considered a total of $N = 3037$ neurons randomly distributed in a 2-dimensional square, with each side measuring 1500 $\mu$m. Details on the model construction, relevant parameters, and their grounding to the literature are outlined in the Methods section, but we briefly summarize the main aspects here.

The main challenge in our computational approach is to represent neural connectivity in a way that integrates single-cell morphology. Following the ideas of previous modeling studies [22, 23], we approach this problem by assuming simplified neuronal shapes paired with synaptic contact probability clouds based on experimental data. Each synthetic neuron is composed of a soma, a dendritic tree with variable size, and an axon. Axons are generated following a biased random walk starting from each neuron's soma (see Fig 1). Whenever there is an intersection between a dendritic tree and an axon, a synapse is established according to a *synaptic contact probability* (SCP). The SCP function quantifies the likelihood of encountering a branch with a spine at a certain radial distance from the soma, denoted as $r$. Consequently, the SCP is influenced by the unique morphological attributes of each genotype (WT, Ts65Dn, and TgDyrk1A).

### Synaptic contact probability from morphological data

To model the impact of neuromorphological alterations on the neural network topology in healthy and DS conditions we analyzed 18 individual pyramidal neurons from WT, Ts65Dn,

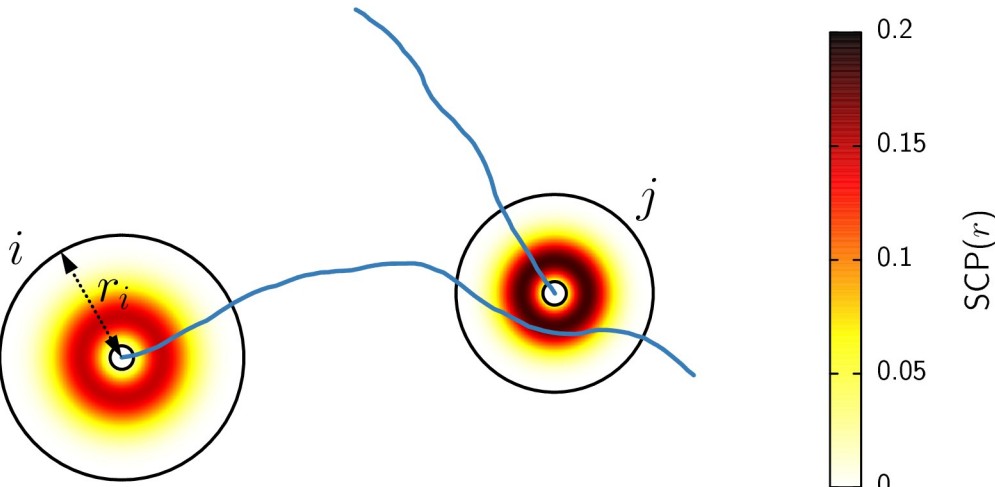

**Fig 1. Schematic representation of the neural network topology generation.** Small and large black circles represent the neurons' soma and dendritic tree, respectively. The color gradient of the dendritic tree corresponds to the synaptic contact probability of each of the two neurons. Blue curves depict the axons, grown according to a biased random walk. Since the axon of neuron *i* overlaps with the dendritic tree of neuron *j*, a synapse might be established. The strength of the synapse depends on the length of the overlap and the value of the SCP along the coincident sites.

and TgDyrk1A animal models (6 neurons for each genotype, see Methods) to calculate SCP based on single-cell morphology data.

Reconstructions of the analyzed neurons are displayed in Fig 2(a)–2(c). For each cell, we obtained the dendritic branching using a Sholl analysis [24]. The Sholl intersection profile is obtained by counting the number of dendritic branches at a given distance from the soma and is a key measure of dendritic complexity. Fig 2(d) shows the average number of intersections as a function of the distance from the soma $r$ for each genotype. Neurons corresponding to the pathological conditions display significantly less branch density and shorter dendritic trees than the control condition (WT). Nonetheless, the shape of the branch density distribution, e.g. how branches are distributed or clustered in a particular area, exhibits a consistent similarity among the three cases, with variations primarily attributable to scaling factors. Using the WT case as the reference, we performed a non-linear fitting of the averaged Sholl intersection profile using the function

$$\mathrm{BD}(r) = ax \exp(-bx^4) \ . \tag{1}$$

The choice of this function allows for an appropriate fit of the data with only two free parameters. The black curve in Fig 2(d) depicts the outcome of the fitting, with the resulting function parameters detailed in Table 1.

Next, we consider the dependency of spine density on the distance from the soma. Here we use data previously published in [2, 3] (see Fig 2(e)). Again, the maximal spine density remains comparable across trisomic, transgenic, and WT genotypes, but is influenced by the shorter dendritic trees in pathological conditions. We fit the WT spine density distribution using a 6th

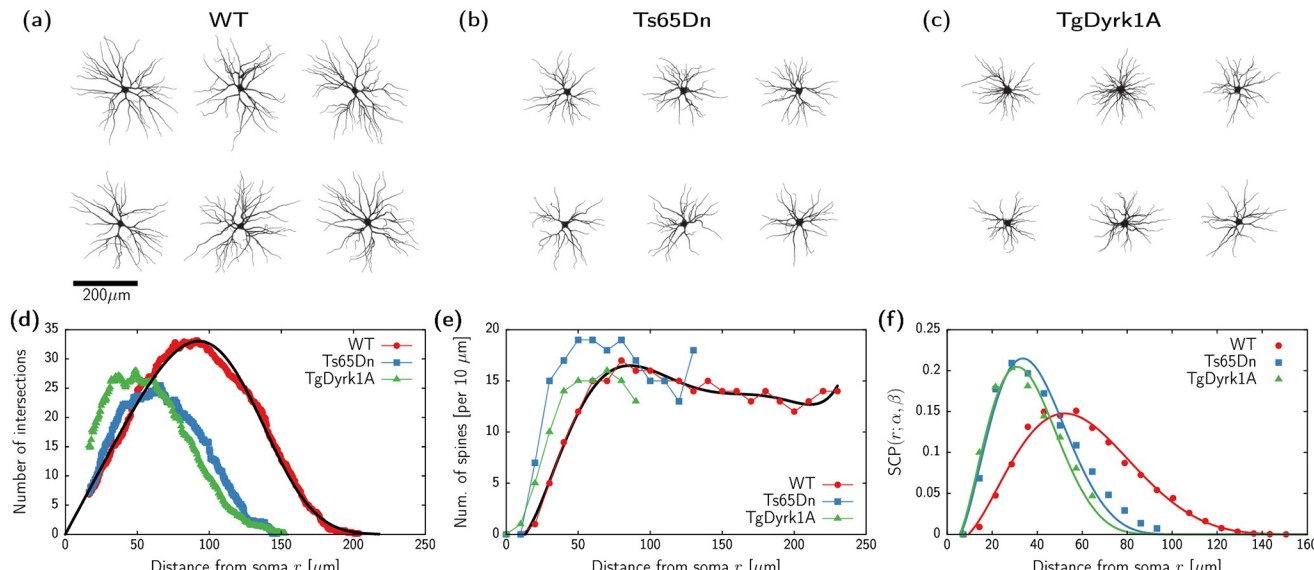

**Fig 2. Neuromorphological data and corresponding models.** (a-c) Reconstructed neurons for each of the three animal models. (d) Sholl intersectional profile (branch pattern complexity) for the three different genotypes (WT, Ts65Dn, and TgDyrk1A). Each dataset corresponds to the average reconstruction of 6 different neurons. Black curve corresponds to fitting Eq (1) to the WT data (see Table 1). (e) Spine density for each different genotype as published in previous literature. The spine numbers are per 10 $\mu$m, thus the distributions are divided by 10 in the fitting procedure. Black curve corresponds to fitting Eq (2) to the WT data (see Table 1). (f) Resulting synaptic contact probability function. Circles, squares, and triangles obtained from the data presented in panels (d) and (e), continuous curves obtained from the nonlinear fitting of Eq (4) using the rescaling parameters $\alpha$ and $\beta$ (see Table 2).

**Table 1. Results of the nonlinear least-squares fitting of BD($r$) and SD($r$).**

| Parameter | Value |
|-----------|-------|
| $a$ | 0.458 |
| $b$ | $3.39 \cdot 10^{-9}$ |
| $c_0$ | 0.0432 |
| $c_1$ | $-0.258$ |
| $c_2$ | 0.0244 |
| $c_3$ | $-4.20 \cdot 10^{-4}$ |
| $c_4$ | $3.12 \cdot 10^{-6}$ |
| $c_5$ | $-1.08 \cdot 10^{-8}$ |
| $c_6$ | $1.44 \cdot 10^{-11}$ |

degree polynomial

$$\mathrm{SD}(r) = \sum_{j=0}^{6} c_j x^j \ . \tag{2}$$

Table 1 contains the parameter values obtained from the fitting, and the black curve in Fig 2(e) shows the resulting function.

The product of the BD and SD functions provides the average spine density at a certain distance from the soma. These functions assume straight dendrites, whereas these are actually irregularly shaped in nature. Indeed, while the largest distance between a spine and the soma in the WT case is 218 $\mu$m, using a convex polygon fitting of reconstructed WT neurons, we found an average mean dendritic tree radius of $\overline{R}_{\mathrm{WT}} = 156.30\,\mu$m (see Methods). To account for the actual shape of the dendritic tree, we rescale the radius in the BD and SD functions by a factor $\gamma = 218/156.30$. Finally, we divide by $2\pi r$ to account for the circular shape of the dendritic tree.

Altogether, for a typical WT neuron, the probability of finding a spine at a distance $r$ from the soma is given by

$$\mathrm{SCP}(r) = \frac{\mathrm{BD}(\gamma r)\mathrm{SD}(\gamma r)}{2\pi r} \ . \tag{3}$$

This function is depicted in Fig 2(f) (see red continuous curve), together with the morphological data (see red circles).

In order to obtain a SCP distribution for Ts65Dn and TgDyrk1A, we exploit the fact that their spine and branch densities in Fig 2(d) and 2(e) follow a similar shape to the WT case up to scaling factors. Therefore, we consider the following generalization of the SCP:

$$\mathrm{SCP}(r; \alpha, \beta) = \alpha \frac{\mathrm{BD}\left(\frac{\gamma}{\beta}r\right)\mathrm{SD}\left(\frac{\gamma}{\beta}r\right)}{2\pi r} \ . \tag{4}$$

Here, the parameter $\alpha$ determines an overall scaling of the SCP in comparison to the WT, whereas $\beta = \overline{R}/\overline{R}_{\mathrm{WT}}$ provides the ratio of the mean dendritic tree radius in comparison to WT. We fit the expression in Eq (4) to the data corresponding to Ts65dn and TgDyrk1A (see blue squares and green triangles in Fig 2(c)) with $\alpha$ and $\beta$ as free parameters. Table 2 contains the resulting values of the neuromorphological parameters, and continuous curves in Fig 2(f) display the resulting shape of the SCP.

**Table 2. Parameter values corresponding to the synaptic contact probability obtained from fitting Eq (4) to the data with $\alpha$ and $\beta$ as free parameters.**

| Parameter | WT | Ts65Dn | TgDyrk1A |
|---|---|---|---|
| $\alpha$ | 1.0 | 0.937 | 0.826 |
| $\beta$ | 1.0 | 0.644 | 0.597 |
| $\overline{R} = \beta \overline{R}_{\mathrm{WT}}$ | 156.30 | 100.66 | 93.31 |

The good agreement between the SCP model (Eq (4)) and the morphological data for the three genotypes substantiates the characterization of single neurons' morphology alterations in DS with only two parameters, $\alpha$ and $\beta$.

## Morphology changes impact network connectivity

We generate network topologies corresponding to each of the studied genotypes with the synaptic contact algorithm described in Methods and illustrated in Fig 1. Topology generation parameters are identical for all genotypes (WT, Ts65Dn, and TgDyrk1A) except for $\alpha$ and $\beta$ as described in the previous section (see Table 2).

In order to characterize the main circuitry changes between the three genotypes, we analyze and compare the topologies in terms of network density, degree centrality, and synaptic strength (see Methods for definitions). To account for the variability on different network generations, we create and analyze 10 different networks for each mouse model. Table 3 contains the average quantities of the computed measures for each case, and Fig 3 displays the corresponding degree and weight distributions.

First, in terms of network density, the three genotypes provide sparse topologies, with less than 7% of all possible links present. Moreover, pathological genotypes show a significant decrease of network density with respect to control. The in-degree and out-degree distribution of the networks (see Fig 3(a) and 3(b)) also reflect this reduction of connectivity in the Ts65dn and TgDyrk1A models, with the average degree being almost half of that of the WT in both cases. In spite of the reduced density of TgDyrk1A compared to Ts65Dn, the degree distributions do not show significant changes between the two pathological cases.

Since in our model a presynaptic neuron can establish several contacts with the same postsynaptic neuron, we also take into account how such synaptic strength varies across genotypes (see Fig 3(c)). In this case, the differences between DS models and WT are less prominent, with only TgDyrk1A showing a decrease of average synaptic strength with respect to WT.

Overall, this analysis reflects that morphology changes incur a direct impact on the connectivities generated by our model. These differences are mainly reflected by an important decrease of number of pre and post-synaptic connections established by each neuron in the DS models, with the strength of such connections displaying only mild changes between genotypes. This scenario is consistent across network generations, as indicated by the small standard deviations shown in parenthesis in Table 3 and in the shaded regions of Fig 3.

**Table 3. Global topological measures for each genotype.** Each value indicates the average measure over a population of 10 networks generated with the same genotype parameters $\alpha$ and $\beta$. Values in parenthesis correspond to the sample standard deviation. Precise definitions for each measure are provided in the Methods section. The source code to generate network topologies is openly available at github.com/pclus/neuromorphology.

| Genotype | Network density (%) | Average degree | Average synaptic strength |
|---|---|---|---|
| WT | 6.6 (0.051) | 200 (1.6) | 11 (0.025) |
| Ts65Dn | 4.0 (0.027) | 120 (0.81) | 11 (0.019) |
| TgDyrk1A | 3.6 (0.039) | 110 (1.2) | 9.4 (0.024) |

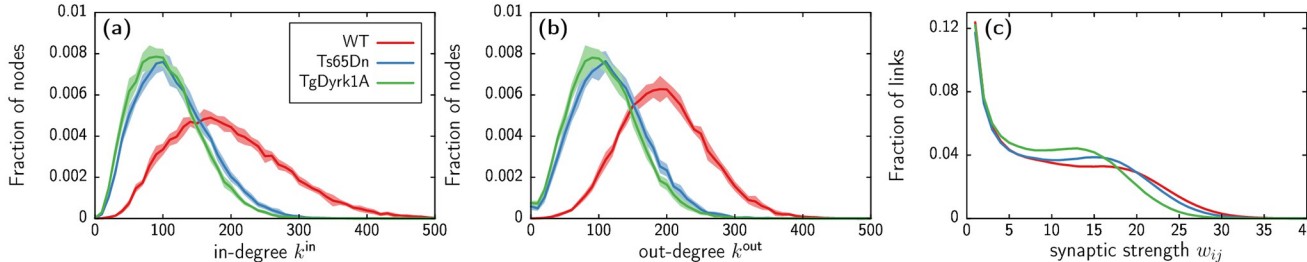

**Fig 3. Topological measures distribution for each genotype.** (a,b) In-degree (panel (a)) and out-degree (panel (b)) distributions corresponding to WT (red), Ts65Dn (blue), and TgDyrk1A (green). (c) Distribution of synaptic strengths, i.e., number of connections between the same two neurons. In all panels, lines correspond to the average of 10 network generations, with shaded regions indicating standard deviation. Standard deviation in panel (c) is too small to be visibly appreciated.

## Simulations recapitulate disrupted gamma activity in pathological conditions

Next, our aim is to investigate how the topological changes induced by morphology alterations affect the network at a functional level. We simulate the dynamics of each neuron using the Izhikevich model with parameters set for regular spiking (pyramidal) and fast-spiking (inter-neurons) [21] (see Methods). Importantly, each neuron in the network receives an independent train of external excitatory inputs following a Poisson shot process with frequency λ. The code to simulate the network is openly available at github.com/pclus/neuromorphology.

Fig 4(a)–4(f) show raster plots and mean-firing rate time series for each genotype obtained from network simulations with λ = 9 kHz. Single unit spike trends are rather irregular due to the three different stochastic sources in the model: randomness of the network generation, external inputs λ, and finite-size effects. Nonetheless, noisy collective oscillations emerge due to the interplay between the fast-spiking interneurons, the excitatory neurons, and the external excitatory input. This onset of gamma rhythmic activity in a noisy environment corresponds to the paradigmatic pyramidal-interneuron network gamma (PING) mechanism [13, 25–27]. S1, S2 and S3 Movies show such dynamic activity at both single-cell and collective levels.

Visual inspection of individual simulations in Fig 4(a)–4(f) already indicate a lack of synchronicity in the Ts65Dn and TgDyrk1A models. In order to properly compare the dynamics of the three cases, we capture the collective activity in each simulation by computing the local field potential (LFP) as the average network firing rate (see Methods). Furthermore, to test the consistency of the results against statistical fluctuations of the topology generation and external input simulation, we use 10 networks for each of the three neuronal genotypes, and each of them is simulated independently 10 times, resulting in a total pool of 100 time series for each parameter set.

Continuous lines in Fig 4(g) show the average power spectra of the LFP corresponding to each animal model for input rate λ = 9 kHz. In all three genotypes, the spectrum shows a clear peak around 40 Hz, with very small deviation across different noise realizations, confirming the robustness of the gamma activity in the model. Nonetheless, the TgDyrk1A and Ts65Dn models display a clear reduction of gamma power compared to the WT case. This decrease in power is paired with a slight decrease in the peak frequency. These results align with empirical observations of reduced gamma activity in prefrontal cortex of TgDyrk1A mice with compared to WT [7].

Further assessment of the dynamical differences between the three genotypes is provided by the analysis of the interspike interval (ISI) distribution. S1(a) and S1(b) Fig show the

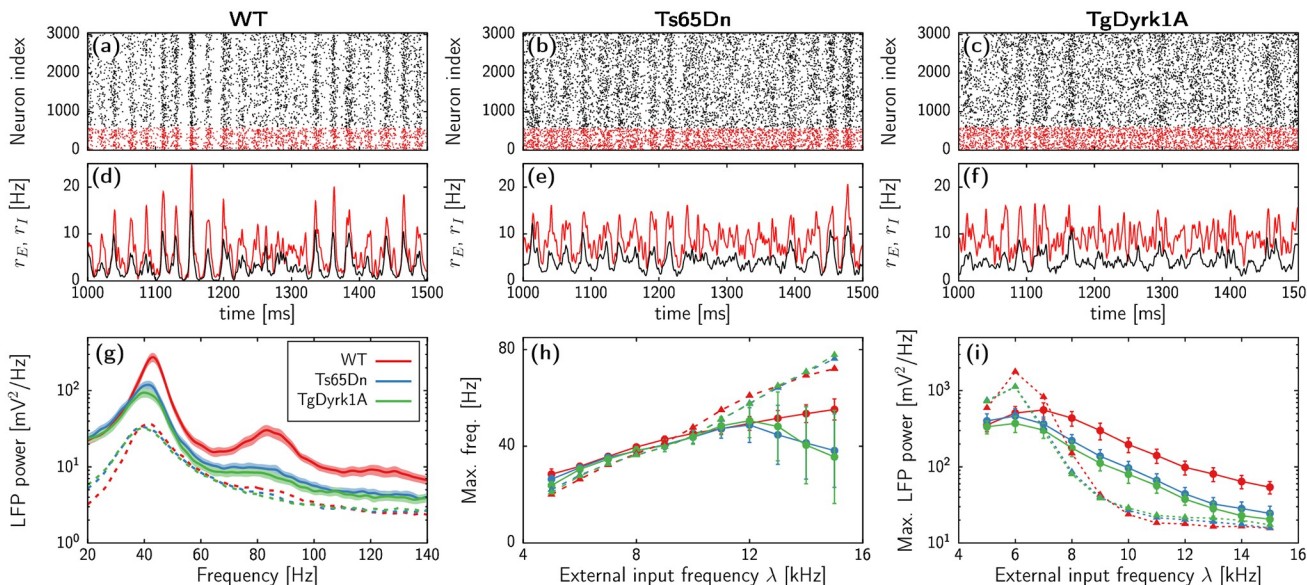

**Fig 4. Neuronal network activity.** (a-c) Raster plots showing the spike times of excitatory (black) and inhibitory (red) neurons for WT (panel (a)), Ts65Dn (panel (b)), and TgDyrk1A (panel (c)). (d-f) Mean firing rate of excitatory (black) and inhibitory (red) neurons corresponding to the raster plots shown in panels (a-c). (g) Power spectrum of LFP signals for an input frequency of λ = 9 spikes/ms. Each curve corresponds to the average of 100 spectra corresponding to 10 independent realizations of the noise for 10 different topologies. Shaded regions indicate the standard deviation among the samples. Red, blue, and green correspond to the morphological parameters of the WT, Ts65Dn, and TgDyrk1A cases, respectively (see Table 2). Dashed lines correspond to simulations with recurrent inhibitory synapses reduced to 0.3 of the original value. (h,i) Location of the peak power in the averaged LFP power spectra (h) and corresponding power (i) obtained for different values of the external firing rate λ. Circles and continuous lines indicate results using the default network parameters. Triangles and dashed lines correspond to reduced recurrent inhibition, as in panel (g). Error bars indicate the standard deviation over the 100 simulations pool.

average interspike interval distribution of each simulation for excitatory and inhibitory neurons respectively. Remarkably, the ISI distributions of pyramidal neurons show a broad profile, with a coefficient of variation around 1.1. Moreover, the three genotypes display a peak around 100 ms, indicating thus a preferred spiking period four times larger than the period of the collective gamma oscillation. On the other hand, inhibitory neurons do show a prominence of spikes around 25 ms, revealing thus the significance of interneuron activity for the emerge of gamma oscillations. The WT ISI distribution of inhibitory neurons also presents a larger peak at shorter periods, corresponding to repetitive firing of interneurons when the excitatory feedback is strong enough.

Next, we investigate the effects of the external firing rate λ on the network dynamics. Fig 4 (h) and 4(i) show the peak frequency and power, respectively, for the three genotypes upon varying λ (solid lines in the two panels). Additionally S3 and S4 Figs show raster plots and mean firing rate activity for individual simulations with λ = 6 and 12 kHz respectively. For all explored values, the three models produce robust oscillatory activity with a peak frequency generally within the gamma range (30–80 Hz). For low values of λ, the network frequency shows similar behavior for all three mouse models, displaying an increase with external input up to λ ≈ 12 kHz (see Fig 4(h)). However, for larger values of external input, the monotonic dependence of the frequency on λ breaks down for the DS models. Such decline in frequency observed in TgDyrk1A and Ts65Dn for large λ values is concomitant with an elevation in variability across network realizations, suggesting a lack of uniformity in generating gamma oscillations in the pathological models.

Regarding the power of the neural oscillations (Fig 4(i)), the WT model demonstrates a notably higher gamma amplitude as compared to the DS models for λ > 7 kHz. These differences in power between genotypes remain mostly unchanged upon increasing λ, although all three models show a reduction of oscillatory coherence as the external input increases. This is consistent with the increase of single neuron spike irregularity, as captured by the average CV displayed in S1(c) and S1(d) Fig. Overall, the scenario remains similar to that displayed in Fig 4(g), and reproduces experimental findings observed in electrophysiology studies of the TgDyrk1A model [7].

## Reduced recurrent inhibition might increase or reduce oscillatory activity

Histological analysis of TgDyrk1A and WT mice cortex shows a significant reduction of the inhibitory synapses acting upon interneurons in the DS model [7]. We expect this to influence the network dynamics, since parvalbumin-positive interneurons are known to modulate gamma activity in the cortex [13, 14, 27]. Moreover, recurrent inhibition, one of the key factors involved in fast collective activity [28], was probably a leading cause of the gamma impairment in the DS model [7] given the reduction of GABAergic contacts among interneurons.

In this section, we test the effect of reduced recurrent inhibition in our model. While *in vivo*, this perturbation of the network balance only occurs for DS animals, we deliberately test the effects of disrupted inhibition in all three genotypes. This allows us to compare the effects of morphology alterations and reduced inhibition separately, both of which coexist in DS animal models.

Dashed lines in Fig 4(g) show the average power spectral density (PSD) obtained from simulations in networks with inhibitory-to-inhibitory synaptic strength reduced to 30% for λ = 9 kHz. While the three genotypes still exhibit a prominent peak around 40 Hz, there is a substantial decrease in activity across all frequency bands when contrasted with the unperturbed models. Significantly, the distinctions in power between WT and DS models vanish, resulting in nearly identical spectra for all three. Raster plots and mean firing rates of individual simulations in S2, S3 and S4 Figs indicate that this reduction of power corresponds to an increased inhibitory activity, which in turns lowers the activity of excitatory neurons.

Once more, we test the soundness of this scenario upon changing λ. Dashed lines and triangles in Fig 4(h) show the peak frequency for the models with weakened recurrent GABAergic contacts. For low values of the external input, the main frequency of oscillation remains close to the unperturbed models. Moreover, disrupted inhibition rescues the drop in gamma frequency of the DS models reported in the previous section. Indeed, with higher values of λ all three genotypes exhibit faster oscillatory activity compared to their respective unperturbed models, with no discernible distinctions between the three genotypes.

The stimulating effect of decreased recurrent inhibition on the oscillatory frequency within the DS models differs from its effects on peak power. Dashed lines and triangles in Fig 4(i) show two different scenarios depending on whether the external input is smaller or larger than λ ≈ 7 kHz. With lower external activity levels, disrupted recurrent inhibition enhances oscillatory power, with the WT model consistently exhibiting greater power than the DS models. Conversely, for higher external activity levels, the power of the three modified models rapidly declines below the levels of the default models, with all three genotypes reaching a plateau for λ > 10 kHz. In this range, no substantial differences in power are discernible among the three genotypes. Furthermore, within this range, the impact of disrupted excitation-inhibition balance on gamma power in the WT appears to be twofold compared to the drop of gamma between the unperturbed WT and DS models. Since frequencies above

40 Hz require $\lambda > 9$ kHZ (Fig 4(h)), these results suggest that reducing recurrent inhibition notably impedes gamma synchronicity, in agreement with [7].

The abrupt reduction in peak power in the disrupted networks corresponds to a transition from a highly synchronized state to a regime in which neurons fire irregularly, while still exhibiting some degree of collective synchronous behavior, mainly through the interneuron population. These two forms of gamma activity have been identified in previous computational studies and are usually referred to as *strong* and *weak* gamma, respectively [29, 30]. The sensitivity of the network rhythmicity on the external input $\lambda$ in the perturbed models highlights the importance of recurrent inhibition to obtain robust gamma rhythms in cortical neural networks.

## Parameter exploration

The unified SCP (Eq (4)) function derived from the morphological data for the three animal models enables the investigation of the network activity generated by hypothetical neuron morphologies. In this context, we explore the influence of the spine density ($\alpha$) and dendritic tree size ($\beta$) on the power and frequency of the gamma rhythm.

Fig 5 shows the outcome of the LFP signal obtained from numerical simulations in network topologies generated with specific values of the scaling parameters for the mean dendritic tree $\beta$ and synaptic contact probability $\alpha$. The peak of the gamma activity is observed at higher frequencies for networks with low values of $\alpha$ and high values of $\beta$. Conversely, the power of such gamma activity becomes larger when both morphological parameters are high. This dual relationship highlights that there is no specific region within the morphological space where both frequency and power can be maximized concurrently. Instead, the emergence of these fast oscillations is driven by the topological features of the networks in a nonlinear manner.

When using the exact values of the animal models in this exploration, it becomes evident that the most prominent difference between the DS and the WT is given by a substantial reduction of the parameter $\beta$. This implies that the size of the dendritic tree exerts a pivotal influence on the gamma abnormalities when compared with the reduction of the SCP parameter $\alpha$. Consequently, our computational results indicate that the gamma impairment in DS models is primarily attributed to the loss of synaptic connections among neurons rather than a decline in the overall strength of these connections. Nonetheless, the parameter exploration

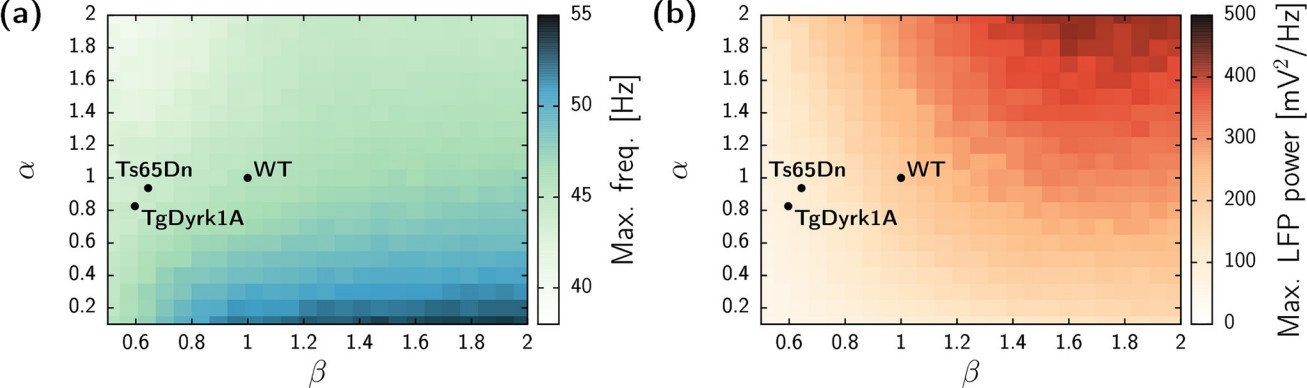

**Fig 5. Network activity dependence on the morphological parameters showing the frequency peak of the gamma rhythm (a) and corresponding LFP power (b).** Results from numerical simulations of networks for external input $\lambda = 10$ kHz. Each pixel of the heatmap corresponds to the average of results obtained with 10 different network topologies, each simulated with 10 different realizations of the noise. Parameters corresponding to the studied genotypes are marked with black circles for reference.

suggests that changes on SCP stronger than those observed in Ts65Dn and TgDyrk1A could also significantly alter gamma oscillations (see also S5 Fig).

## Discussion

In this study, we aimed to explore the interplay between neuromorphological alterations and network dynamics in DS, using a data-driven computational model. Our findings provide compelling evidence that the incorporation of empirical data on dendritic complexity and spine density into the model enables the faithful replication of the reduction in gamma oscillations documented in DS animal models [7]. These results strongly indicate that the neuromorphological changes observed in DS are pivotal contributors to the disruption of gamma activity, providing valuable insights into the underlying mechanisms of network dynamics associated with DS.

An important challenge of this work has been to integrate single-cell morphology data into the generation of the neural network architecture. Few studies include morphology in neural network topology, and the options vary based on the desired biological realism: from sophisticated cloning of reconstructed morphologies paired with touch-detection rules [31–33], to generating random networks based mostly on axo-dendritic overlap [22, 34]. A mid-way approach consists of defining a probability region based on different morphological parameters [23, 35, 36].

Here we took this intermediate route by incorporating a synaptic contact probability (SCP) function into the model developed in [22]. Through an analysis of single-cell morphology data, we devised a quantitative model for the SCP that integrates dendritic complexity and spine density of different mouse models. Remarkably, our findings demonstrate that the three genotypes examined in this study (WT, trisomic, and single-gene transgenic TgDyrk1A) can be accurately represented by a single SCP function, with scaling factors accounting for their distinct characteristics. This result allowed us to explore the broader morphological landscape beyond the confines of specific animal models.

The simulations conducted in the study demonstrate the influence of diminished recurrent inhibition on network dynamics. The reduction of inhibitory connections, particularly those targeting parvalbumin-positive interneurons, previously reported in DS animal models [7], has been proposed as a potential cause for the observed reduction in gamma oscillation. The model's implementation of reduced recurrent inhibition successfully reproduces the effects on gamma activity, underscoring its pivotal role in modulating network dynamics. Notably, perturbing excitation-inhibition balance rescues the drop in frequency in DS models but concurrently diminishes oscillatory power. These results align with previous empirical and modeling studies identifying recurrent inhibition as a key ingredient for the emergence of gamma oscillations [14, 27, 28, 37–39].

Furthermore, the study explores the morphological parameter space and demonstrates that there is no single regime in which both frequency and power of gamma oscillations can be maximized. Moreover, the reduction in dendritic tree size ($\beta$) appears to be a major factor contributing to the gamma abnormalities in DS, while the reduction in synaptic contact probability ($\alpha$) plays a secondary role.

In summary, the data-driven computational model presented in this study successfully integrates neuromorphological alterations observed in DS animal models and reproduces the reduction in gamma oscillations. The findings support the notion that microscopic circuitry abnormalities contribute to the disruption of network dynamics in DS. The model provides a controlled framework to explore the impact of morphology alterations and elucidate their role in network synchronicity.

## Model limitations

Our data-driven computational approach relies on several modeling assumptions that should be taken into account when drawing biological conclusions from the results. One major simplification pertains to the morphological model for network topologies, which simplifies neuronal shapes and may not fully capture the diversity and intricacy of real morphologies. Furthermore, the axon growth model doesn't consider the precise axon terminal locations.

Another aspect is the dynamical model employed for the simulations. Here we considered idealized point neurons, which are simplified models that disregard the influence of detailed structures, such as dendrites and axons, on the dynamics of the cell. While these simplifications facilitate easier analysis and interpretation, they may not fully capture the effects that morphology can have on the system dynamics. To this end, compartmental models could allow the incorporation of specific relations such as the effects of smaller dendritic trees on the synaptic delays.

Despite these limitations, our data-driven computational model is an important milestone in understanding the importance of neuromorphological alterations in neurodevelopmental diseases. Further work should allow us to improve and further validate our assumptions based on empirical data, as well as focus on other brain areas which are known to be also affected in DS, such as the hippocampus.

## Methods

### Data gathering

We used previously published single-neuron 2D tracings of cortical layer II/III pyramidal neurons. Those were traced as part of previous studies of our team, following experimental procedures detailed in [40–42]. Briefly, cells were injected with Lucifer Yellow, immuno-stained with a biotinylated secondary antibody and biotin–horseradish peroxidase complex. Tissue sections were imaged with brightfield microscopy and traced using camera lucida microscope attachment. Specifically, the details for the tracing of WT and Ts65Dn neurons can be found at [2], and those for TgDyrk1A neurons can be found at [3]. We did not find differences in neuronal morphology between the WT strains of trisomic and transgenic mice, and thus we consider WT parameters to be the same for the two DS mouse models, thus allowing comparisons among the three genotypes. All data used in this study is openly available at github.com/pclus/neuromorphology.

### Topology generation

The generation of the network topology has been largely inspired by the model developed by Orlandi et. al. [22]. We included substantial modifications to their proposal to account for specific morphological variants of single neurons. A C code of the resulting algorithm is openly available at github.com/pclus/neuromorphology.

Using a full 3D representation of the cortical layer II/III with realistic neuronal densities is not possible due to computational limitations [22, 43]. Thus we considered a thin 2-dimensional cross-section of the cortical layer with a thickness of a single cell soma, 16 $\mu$m.

The neuronal density of the synthetic circuit is set to 1350 neurons/mm$^2$, obtained by multiplying the neuronal density in the 3D layer by the width of the thin layer modeled. Based on this assumption, we placed randomly $N = 3037$ neurons in a 2-dimensional square with a side length of 1.5 mm. To mitigate the effects of imposing a too-small spatial domain, we provided

the circuit with periodic boundary conditions. The spatial resolution of the *in silico* layer is 1 $\mu$m.

In the model, each neuron has three components: the soma, the dendritic tree, and the axon (see Fig 1 for a schematic representation). All neurons have identical soma, which are modeled as circles of radius $R_S$ = 16 $\mu$m. The center of each neuron's soma is randomly distributed on the 2-dimensional layer, and overlapping somas are not allowed. The dendritic tree of each neuron is also modeled as a circle, but in this case, we considered variability among neurons. The radius of each neuron's dendritic tree is a random number drawn from a Gaussian distribution with mean $\overline{R}$ and standard deviation $\sigma$ = 40. The mean dendritic tree radius $\overline{R}$ is one of the main control parameters of the study. As explained in the Results section, we use the WT mean dendritic tree radius $\overline{R}_{WT}$ as a reference. We measured this quantity by calculating the 2D convex hull of the reconstructed WT dendritic trees. Specifically, we used the "boundary" function in MATLAB with shrink factor $s$ = 0. To obtain the mean radius we assumed that the area obtained with the convex hull forms a circle for each tree.

The axon is modeled as a biased random walk starting from the center of the soma with a random direction. After it has grown 10 $\mu$m, it modifies direction by $\theta$ degrees, where $\theta$ is a random variable chosen from a Gaussian distribution with zero mean and standard deviation $\sigma$ = $\pi/30$ radians. The total length of each axon is obtained from a Rayleigh distribution with mean $\overline{l} = 500\,\mu$m. This mean axon length has been chosen to simulate only local horizontal connections (given by the local axonal tree in layers II/III), following experimental observations in the mouse M2 cortical layer II/III [43], and disregarding horizontal patchy connections, connections between cortical layers, and interhemispheric projections.

If the axon of neuron $i$ overlaps the dendritic tree (but not the soma) of neuron $j$, then a synapse is established with a probability $p$ that depends on the distance $r$ from the overlap point to the neuron's $j$ soma. Such dependency is given by the function $p$ = SCP($r;\alpha, \beta$) where $\alpha$ and $\beta$ are parameters that depend on the morphological variables. In particular, $\alpha$ is the ratio between the synaptic contact probability of each mouse model and the wild-type (WT) value, and $\beta$ is the ratio between the dendritic tree radius of each mouse model and WT, i.e., $\beta = \overline{R}/R_{WT}$ where $\overline{R}_{WT} = 156.30\,\mu$m is the mean dendritic tree radius of a WT neuron. The derivation of the synaptic contact probability function SCP($r$) and the values of $\alpha$ and $\beta$ for the different animal models are detailed in the Results section.

Each grid square presenting an overlap between an axon and a dendritic tree can generate a synapse, thus each pair of neurons might have multiple, in some cases several, synapses. In other words, the resulting architecture of connections among neurons is a directed weighted network given by the weight matrix $W$ = ($w_{ij}$). Autapses are not allowed.

## Topology measures

We analyze the network topologies generated by our model using standard measures of complex network analysis:

1. **Network density:** Proportion of links present in the network with respect to the total number of possible links:

$$\rho = \frac{1}{N(N-1)} \sum_{i,j=1}^{N} a_{ij} \tag{5}$$

where $A = (a_{ij})$ is the network adjacency matrix, i.e.,

$$a_{ij} = \begin{cases} 1 & \text{if} \quad w_{ij} \neq 0 \\ 0 & \text{if} \quad w_{ij} = 0 \ . \end{cases} \tag{6}$$

2. **In and out-degree:** For each node $i$, the in-degree $k_i^{\text{in}}$ is the number of pre-synaptic neurons with a contact on $i$. The out-degree $k_i^{\text{out}}$ corresponds to the number of post-synaptic neurons connected by $i$. They are computed as

$$k_i^{\text{in}} = \sum_{l=1}^{N} a_{il} \quad \text{and} \quad k_i^{\text{out}} = \sum_{l=1}^{N} a_{li} \ . \tag{7}$$

3. **Synaptic strength:** Number of established connections between the same two neurons, $w_{ij}$.

## Neuronal dynamics

Several models for spiking neuron dynamics exist in the literature. Here we use the model proposed in [21], due to its apt trade-off between dynamical richness and computational efficiency. The C code used to simulate the network dynamics is openly available at github.com/pclus/neuromorphology. We consider a network composed of pyramidal and fast-spiking interneurons only. Since the topology generation algorithm does not differentiate between the two types, each neuron is set as either excitatory or inhibitiory randomly at the beginning of each simulation with a 80%-20% proportion [26].

The dynamics of the $i$-th neuron in the network is ruled by the following ordinary differential equations,

$$\dot{v}_i = 0.04v_i^2 + 5v_i + 140 - u_i + I_i^{\text{AMPA}} + I_i^{\text{GABA}} + I_i^{\text{ext}} \tag{8}$$

$$\dot{u}_i = a(bv_i - u_i) \tag{9}$$

where $v_i$ is the membrane voltage potential, and $u_i$ is a recovery variable accounting for the dynamics of the different ion channels. When the voltage of neuron $i$ reaches a threshold of $v^{(\text{thr})} = 30$, the neuron emits a spike and the voltage is reset to $v_i \leftarrow c$, whereas $u_i \leftarrow u_i + d$. Upon tuning the system parameters $a$, $b$, and $d$, one can obtain different dynamical behaviors for each neuron. Here we focus on Regular Spiking (RS) for pyramidal neurons and Fast Spiking (FS) for interneurons [21]. Table 4 indicates the numerical value for the parameters corresponding to each neuron type.

The voltage of each neuron is influenced by the synaptic inputs coming from the excitatory neurons of the network, $I_i^{\text{AMPA}}$, inhibitory neurons $I_i^{\text{GABA}}$, and glutamatergic inputs coming

**Table 4. Values of the system parameter to display regular spiking for excitatory neurons, and fast-spiking for inhibitory neurons.**

| Parameter | Pyramidal (RS) | Interneuron (FS) |
|:---:|:---:|:---:|
| $a$ | 0.02 | 0.1 |
| $b$ | 0.2 | 0.2 |
| $c$ | -65 | -65 |
| $d$ | 8 | 2 |

from outside the network, $I^{\text{ext}}$. In all cases, these inputs have the form

$$I_i^{\text{syn}} = g_i^{\text{syn}}(t - \tau_0)(v_R^{\text{syn}} - v_i) \, . \tag{10}$$

with $v_R^{\text{syn}}$ being a reversal potential and $g_i^{\text{syn}}$ is a time-dependent conductance, and $\tau_0 = 1$ ms is a fixed synaptic delay. Upon receiving a spike, the neurotransmitter-activated ion channels open and close following an exponential decay. For the recurrent AMPA and GABA connectivities this reads

$$g_i^{\text{syn}}(t) = g^{\text{syn}} \sum_{m=1}^{N} w_{im} \sum_{k} e^{(t - t_m^{(k)})/\tau^{\text{syn}}} H(t - t_m^{(k)}) \tag{11}$$

where $\mathbf{W} = (w_{ij})$ is the weight matrix of the network, $t_m^{(k)}$ is the time at which neuron $m$ emitted its $k$-th spike, $H$ is the Heaviside step function, $\tau_{\text{syn}}$ is the decay time, $g^{\text{syn}}$ is the strength of the synapse. The incoming signals from outside of the modeled layer consist of excitatory exponential pulses only

$$g_i^{\text{ext}}(t) = g^{\text{ext}} \sum_{k} e^{(t - t^{(k)})/\tau^{\text{ext}}} H(t - t^{(k)}).$$

These inputs account for post-synaptic potentials from other cortical layers, as well as other brain regions projecting to layer 2/3. The spiking times $t^{(k)}$ are drawn from a Poissonian shot process with frequency $\lambda$. We use $\lambda$ as the main control parameter to test the oscillatory response of the network. Values for $g^{\text{syn}}$, $\tau^{\text{syn}}$, and $v_R^{\text{syn}}$ for the three synaptic types are given in Table 5.

In order to capture the network activity, we model the local field potential (LFP) as the network average firing rate (with time bins of 0.1 ms):

$$\text{LFP}(t) = \frac{1}{N} \sum_{m=1}^{N} \sum_{k} \left[ H(t - t_m^{(k)}) - H(t - t_m^{(k)} - 0.1) \right] \, . \tag{12}$$

We opted to use the firing rate as a proxy to capture the actual network activity in our model. Other options, which are based on the AMPA and GABA currents of the network, could be susceptible to parameter changes such as when we model disrupted networks by reducing $g^{\text{GABA}}$ targeting inhibitory neurons. Power Spectral Density of LFP time series have been computed using the GNU Scientific Library [44] by first applying a fast Fourier transform algorithm and then reducing the noise in the spectra through a Gaussian filter.

The firing activity of individual neurons and their regularity can be assessed by the interspike interval (ISI) distribution and the corresponding coefficient of variation (CV). For each neuron $i$ we compute the time between consecutive spikes as $\text{ISI}_i^k = t_i^{(k+1)} - t_i^{(k)}$. For periodically firing neurons, the ISI distribution approaches a delta function located at the period. On the other hand, a broad ISI distribution reflects more complex, irregular, spiking patterns. To characterize the regularity of the firing of each population type, we compute the ISI coefficient

**Table 5. Values of the synaptic strength $g^{\text{syn}}$, decay times $\tau^{\text{syn}}$, and reversal potential $v_R^{\text{syn}}$ for the ifferent neurotransmitters.**

|  | $g^{\text{syn}}$ | $\tau^{\text{syn}}$ | $v_R^{\text{syn}}$ |
|---|---|---|---|
| AMPA | 0.006 mS | 2 ms | 0 |
| GABA | 0.720 mS | 4 ms | -70 |
| External AMPA | 0.008 mS | 2 ms | 0 |

of variation as CV = $\sigma/\mu$, where $\mu$ and $\sigma$ correspond to the mean and standard deviation of $\text{ISI}_i^k$ respectively. We perform this analysis for pyramidal and inhibitory neurons separately.

## Supporting information

**S1 Fig. Interspike interval and firing regularity for different network morphologies.** (a,b) Interspike interval (ISI) distribution corresponding to pyramidal neurons (a) and inhibitiory neurons (b) for $\lambda$ = 9 kHz. Each curve corresponds to the average of 100 histograms corresponding to 10 independent realizations of the noise for 10 different topologies. Shaded regions indicate the standard deviation among the samples. Red, blue, and green correspond to the morphological parameters of the WT, Ts65Dn, and TgDyrk1A cases, respectively. Dashed lines correspond to simulations with recurrent inhibitory synapses reduced to 0.3 of the original value. (c-f) Coefficient of variation (CV) of the ISI distribution of pyramidal neurons (c) and inhibitory neurons (d) for different values of external input $\lambda$. Symbols correspond to the average CV of 100 ISI distributions, and errorbars indicate the respective standard deviation. Panels (c) and (d) correspond to the default network parameters, whereas panels (e) and (f) correspond to simulations with recurrent inhibition reduced to 0.3 of the original value.
(PDF)

**S2 Fig. Gamma activity dynamics for an external input $\lambda$ = 9 kHz.** Raster plots and mean firing rate of the pyramidal (black) and inhibitory interneurons (red) for the WT (panels (a) and (d)), Ts65Dn (panels (b) and (e))), and TgDyrk1A (panels (c) and (f)) morphologies. Panels (a-c) correspond to simulations with unperturbed recurrent inhibition (same as in Fig 4(a)–4(f)), and panels (d-f) correspond to recurrent inhibitory synapses reduced to 0.3 of the original value.
(PDF)

**S3 Fig. Gamma activity dynamics for an external input $\lambda$ = 6 kHz.** Raster plots and mean firing rate of the pyramidal (black) and inhibitory interneurons (red) for the WT (panels (a) and (d)), Ts65Dn (panels (b) and (e))), and TgDyrk1A (panels (c) and (f)) morphologies. Panels (a-c) correspond to simulations with unperturbed recurrent inhibition, and panels (d-f) correspond to recurrent inhibitory synapses reduced to 0.3 of the original value.
(PDF)

**S4 Fig. Gamma activity dynamics for an external input $\lambda$ = 12 kHz.** Raster plots and mean firing rate of the pyramidal (black) and inhibitory interneurons (red) for the WT (panels (a) and (d)), Ts65Dn (panels (b) and (e))), and TgDyrk1A (panels (c) and (f)) morphologies. Panels (a-c) correspond to simulations with unperturbed recurrent inhibition, and panels (d-f) correspond to recurrent inhibitory synapses reduced to 0.3 of the original value.
(PDF)

**S5 Fig. Gamma activity dynamics corresponding to selected fabricated morphologies.** (a-d) Raster plots and mean firing rate of the pyramidal (black) and inhibitory interneurons (red) for network topologies generated with different values of the SCP scaling parameter $\alpha$ and the scaling of the mean dendritic tree size with respect to WT $\beta$. Rest of the parameters as in Fig 5. (e,f) Average power spectra of the LFP signal corresponding to Fig 5 for specific values of $\alpha$ and $\beta$.
(PDF)

**S1 Movie. Simulation of the model corresponding to WT morphology.** Top: spatial representation of the cortex slice model. Each circle represents a neuron, whose spikes are marked

with red (excitatory neurons) and blue (inhibitory neurons) for a short time interval. Bottom: Time series of the average firing rate of excitatory (red) and inhibitory (blue) neurons. Simulation parameters as in Fig 4(a) ($\lambda = 9$ kHz).
(MP4)

**S2 Movie. Simulation of the model corresponding to Ts65Dn morphology.** Top: spatial representation of the cortex slice model. Each circle represents a neuron, whose spikes are marked with red (excitatory neurons) and blue (inhibitory neurons) for a short time interval. Bottom: Time series of the average firing rate of excitatory (red) and inhibitory (blue) neurons. Simulation parameters as in Fig 4(a) ($\lambda = 9$ kHz).
(MP4)

**S3 Movie. Simulation of the model corresponding to TgDyrk1A morphology.** Top: spatial representation of the cortex slice model. Each circle represents a neuron, whose spikes are marked with red (excitatory neurons) and blue (inhibitory neurons) for a short time interval. Bottom: Time series of the average firing rate of excitatory (red) and inhibitory (blue) neurons. Simulation parameters as in Fig 4(a) ($\lambda = 9$ kHz).
(MP4)

## Author Contributions

**Conceptualization:** Pau Clusella, Linus Manubens-Gil, Jordi Garcia-Ojalvo, Mara Dierssen.

**Data curation:** Linus Manubens-Gil, Mara Dierssen.

**Formal analysis:** Pau Clusella, Linus Manubens-Gil.

**Funding acquisition:** Linus Manubens-Gil, Jordi Garcia-Ojalvo, Mara Dierssen.

**Investigation:** Pau Clusella, Linus Manubens-Gil, Jordi Garcia-Ojalvo, Mara Dierssen.

**Methodology:** Pau Clusella, Linus Manubens-Gil.

**Resources:** Linus Manubens-Gil, Jordi Garcia-Ojalvo, Mara Dierssen.

**Software:** Pau Clusella, Linus Manubens-Gil.

**Supervision:** Jordi Garcia-Ojalvo, Mara Dierssen.

**Validation:** Pau Clusella, Linus Manubens-Gil.

**Visualization:** Pau Clusella, Linus Manubens-Gil.

**Writing – original draft:** Pau Clusella.

**Writing – review & editing:** Pau Clusella, Linus Manubens-Gil, Jordi Garcia-Ojalvo, Mara Dierssen.

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
