## [Decision Letter · Decision Letter 0]

2 Apr 2024

Dear Dr Clusella,

Thank you very much for submitting your manuscript "Modeling the impact of neuromorphological alterations in Down syndrome on fast neural oscillations" for consideration at PLOS Computational Biology. As with all papers reviewed by the journal, your manuscript was reviewed by members of the editorial board and by several independent reviewers. The reviewers appreciated the attention to an important topic. Based on the reviews, we are likely to accept this manuscript for publication, providing that you modify the manuscript according to the review recommendations.

In particular, some further characterisation of the dynamics of these network simulations is needed; and the authors should confirm whether variations in synaptic contact probability could also affect gamma oscillations for a fixed dendritic tree size.

Sincerely,

Daniel Bush

Academic Editor

PLOS Computational Biology

Daniele Marinazzo

Section Editor

PLOS Computational Biology

Reviewer's Responses to Questions

**Comments to the Authors:**

Reviewer #1: In this manuscript the authors propose a computational model reproducing the dynamics of a cross-sectional layer of the mouse motor cortex. In particular they design the network connectivity such that it accounts both for the mean dendritic radius and the spine density of the single neurons, starting from the data of three mouse models: wild-type, transgenic and trisomic. The idea underlying the manuscript is to characterize the effects in the emergent dynamics of morphological alterations present in neurons in case of cognitive disorders, like Down syndrome (DS). In particular they analyse the emergence of gamma activity and how this activity diminishes in DS models. The findings support the notion that microscopic circuitry abnormalities contribute to the disruption of network dynamics in DS.

The manuscript is well written and interesting and I appreciate the attempt to fill a gap trying to understand the relationship between neromorphological alterations and impaired brain function.

However, I think the manuscript needs to be improved in order to be published, because a characterization of the dynamics emergent in the network (in terms of raster plots and time-behavior of macroscopic variables) is completely missing.

In the following there is a detailed list of my remarks and questions.

1. Since the authors propose a novel computational model, I think it is necessary to characterize the dynamics emergent in the simulated network in more detail. I understand that such characterization may distract the reader from the main message, but such analysis can be reported mainly in the Methods section or in some Supplementary figures, without impacting on the main results.

Since the authors are using a network composed of almost 3000 neurons, where are present both regular spiking pyramidal neurons and fast spiking interneurons, I suggest to present separated results for the inhibitory and for the excitatory neurons. Please present a raster plot in time of both inhibitory and excitatory neurons for different parameters sets.

Are there some differences for the 3 mouse model cases?

Which are the dynamical regimes emerging in this context? Is gamma activity linked to a periodic oscillation regime?

Do the dynamical regimes change when reducing the recurrent inhibitory synapses? or when chancing alpha and beta parameters?

Moreover, please report the LFP in time (or spectrograms of the LFP in time) for the inhibitory and excitatory populations and calculate the coefficient of variation (CV) of the network for different set of parameters.

2. Which is the biological reason underlying the choice of external input frequency lambda around 10 kHz? What does the external current account for? Does it take into account the signals from other cortical layers? Please explain.

3. In the Methods section w_j is used to identify the recovery variable, while w_{i,j} are the entries of the connectivity matrix. I would suggest not to use the same letter for different variables as it is now.

4. Could you please provide some more details on the connectivity matrix used i the model? Which is the average

connectivity in the network? Do excitatory and inhibitory neurons have the same average connectivity?

How much does it vary among the different trials?

Reviewer #2: Clusella et al report about the link between changes in morphology and the disruption of gamma oscillations in DS. For this they developed a cortical-layer structural model to mimic the synaptic connectivity of a neural network of cross section of layer 2/3 of the motor cortex. The model is made of point neurons, including pyramidal and fast-spiking interneurons, and based on the empirical data on dendritic complexity and spine density obtained from wild-type (WT), transgenic (TgDyrk1A), and trisomic mice (Ts65Dn). The authors demonstrated that their model based on morphological changes replicates the disruption of gamma activity observed in the models. The model is based on a unique synaptic contact probability and can be used to reflect the 3 genetic conditions explored here. According to the model, the authors proposed that the reduction in the dendritic tress size is the main factor contributing to the gamma abnormalities in DS while the reduction in synaptic contact probability may have a second role.

It would have been of interest to show if the variation in the synaptic contact probability can affect the gamma oscillation for a fixed dendritic tree size.

Miscellaneous

- Wild-type individuals should be written as “wt” not “WT”

- The gene dual-specificity tyrosine phosphorylation regulated kinase 1a gene should be written according to the international nomenclature Dyrk1a not Dyrk1A

- The full name of the models used in the manuscript should be given and not only short internal labels like TgDyrk1A and Ts65Dn with reference for the original source

**Have the authors made all data and (if applicable) computational code underlying the findings in their manuscript fully available?**

Reviewer #1: Yes

Reviewer #2: Yes

PLOS authors have the option to publish the peer review history of their article (what does this mean?). If published, this will include your full peer review and any attached files.

Reviewer #1: No

Reviewer #2: **Yes: **Herault

Figure Files:

Data Requirements:

Reproducibility:

References:

---

## [Decision Letter · Decision Letter 1]

18 Jun 2024

Dear Dr Clusella,

We are pleased to inform you that your manuscript 'Modeling the impact of neuromorphological alterations in Down syndrome on fast neural oscillations' has been provisionally accepted for publication in PLOS Computational Biology.

Best regards,

Daniel Bush

Academic Editor

PLOS Computational Biology

Daniele Marinazzo

Section Editor

PLOS Computational Biology

Reviewer's Responses to Questions

**Comments to the Authors:**

Reviewer #1: The authors have addressed all my questions.

**Have the authors made all data and (if applicable) computational code underlying the findings in their manuscript fully available?**

Reviewer #1: Yes

PLOS authors have the option to publish the peer review history of their article (what does this mean?). If published, this will include your full peer review and any attached files.

Reviewer #1: No

---

## [Editor Report · Acceptance letter]

29 Jun 2024

PCOMPBIOL-D-24-00075R1 

Modeling the impact of neuromorphological alterations in Down syndrome on fast neural oscillations

Dear Dr Clusella,

I am pleased to inform you that your manuscript has been formally accepted for publication in PLOS Computational Biology. Your manuscript is now with our production department and you will be notified of the publication date in due course.

With kind regards,

Zsofia Freund
